

# Responses of surface ozone air quality to anthropogenic nitrogen deposition in the Northern Hemisphere

Yuanhong Zhao[1], Lin Zhang[1], Amos P. K. Tai[2], Youfan Chen[1], Yuepeng Pan[3]

[1] Laboratory for Climate and Ocean-Atmosphere Sciences, Department of Atmospheric and Oceanic

Sciences, School of Physics, Peking University, Beijing 100871, China

[2] Earth System Science Programme and Graduate Division of Earth and Atmospheric Sciences, Faculty

of Science, The Chinese University of Hong Kong, Hong Kong SAR, China

[3] State Key Laboratory of Atmospheric Boundary Layer Physics and Atmospheric Chemistry (LAPC),

Institute of Atmospheric Physics, Chinese Academy of Sciences, Beijing 100029, China

*Correspondence to*: Lin Zhang (zhanglg@pku.edu.cn)

**Abstract.** Human activities have substantially increased atmospheric deposition of reactive nitrogen to

the Earth's surface, inducing unintentional effects on ecosystems with complex environmental and

climate consequences. One consequence remaining unexplored is how surface air quality might respond

to the enhanced nitrogen deposition through surface-atmosphere exchange. Here we combine a

chemical transport model (GEOS-Chem) and a global land model (Community Land Model) to address

this issue with a focus on ozone pollution in the Northern Hemisphere. We consider three processes that

are important for surface ozone and can be perturbed by addition of atmospheric deposited nitrogen,

namely, emissions of biogenic volatile organic compounds (VOCs), ozone dry deposition, and soil

nitrogen oxide ($NO_x$) emissions. We find that present-day anthropogenic nitrogen deposition (65 Tg N

$a^{-1}$ to the land), through enhancing plant growth (represented as increases in vegetation leaf area index

(LAI) in the model), could increase surface ozone from increased biogenic VOC emissions (e.g., a 6.6

Tg increase in isoprene emission), but could also decrease ozone due to higher ozone dry deposition



velocities (up to 0.02-0.04 cm s$^{-1}$ increases). Meanwhile, deposited anthropogenic nitrogen to soil enhances soil NO$_x$ emissions. The overall effect on summer mean surface ozone concentrations show general increases over the globe (up to 1.5-2.3 ppbv over the western US and South Asia), except for some regions with high anthropogenic NO$_x$ emissions (0.5-1.0 ppbv decreases over the eastern US, Western Europe, and North China). We compare the surface ozone changes with those driven by the past 20-year climate and historical land use changes. We find that the impacts from anthropogenic nitrogen deposition can be comparable to the climate and land use driven surface ozone changes at regional scales, and partly offset the surface ozone reductions due to land use changes reported in previous studies. Our study emphasizes the complexity of biosphere-atmosphere interactions, which can have important implications for future air quality prediction.

## 1 Introduction

Reactive nitrogen, in the forms of reduced (NH$_x$) and oxidized nitrogen (NO$_y$), is an essential nutrient to the biosphere. Without human influence, reactive nitrogen is mainly fixed from inert nitrogen gas (N$_2$) through natural biological fixation, lightning, and wildfires (Galloway et al., 2004; Fowler et al., 2013). Human activities such as urbanization, industrialization, and agricultural development have led to the emissions of large amounts of reactive nitrogen in the forms of nitrogen oxides (NO$_x$ = NO + NO$_2$) and ammonia (NH$_3$) since the preindustrial period. Their removal via atmospheric deposition has increased by more than a factor of three from the preindustrial era to the early 2000s, and become an important source of reactive nitrogen to terrestrial and oceanic ecosystems (Galloway et al., 2004; Liu et al., 2013; Zhao et al., 2017).

Assessing the consequences of atmospheric nitrogen deposition requires a deep understanding of the interactions and feedbacks within different components of the Earth system including the biosphere and the atmosphere. There is evidence that enhanced atmospheric nitrogen deposition has led to negative





effects such as soil acidification (Stevens et al., 2009; Lu et al., 2014), eutrophication (Rodríguez et al., 2006), and loss of biomass diversity (Baron et al., 2014). Atmospheric nitrogen deposition has also been shown to increase carbon storage in terrestrial and oceanic ecosystems, but the resulting climate benefits can be largely offset by increased emissions of nitrous oxide ($N_2O$), a major greenhouse gas, as

a byproduct of enhanced microbial nitrification and denitrification in soils (Duce et al., 2008; Zaehle et al., 2011; Bala et al., 2013). Previous studies mainly focused on the land-atmosphere exchange of long-lived greenhouse gases including $CO_2$, $N_2O$, and $CH_4$ (Liu et al., 2009; Zaehle et al., 2011) and their effects on climate. Very few studies have explored how ecosystem-mediated feedbacks through atmospheric chemistry influence air quality. Here we will present such a study that investigates how

human-induced atmospheric nitrogen deposition may affect atmospheric composition and air quality via modifying ecosystem structure in terms of foliage density, with a focus on surface ozone pollution.

Near-surface ozone is a harmful air pollutant that results in detrimental effects on human health and vegetation (Bates, 2005; Jerrett et al., 2009; Avnery et al., 2011). It is mainly formed in the troposphere

by photochemical oxidation of carbon monoxide (CO) and VOCs in the presence of $NO_x$. Tropospheric ozone burden has been more than doubled since preindustrial times, mainly driven by rising anthropogenic emissions of ozone precursors ($NO_x$, CO, and VOCs) and the recent equatorward shift of emission patterns (Young et al., 2013; Zhang et al., 2016). Ozone impact on plant growth is mainly affected through its stomatal uptake, and has been shown to severely damage forest, grassland and

agricultural productivity (Ainsworth et al., 2012). Ozone damage impedes various foliage physiological functions including photosynthesis and stomatal conductance, with ramifications not only for ecosystem health but also for climate (Fowler et al., 2009; Matyssek et al., 2010; Yue et al., 2014; 2016; Sadiq et al., 2017). Major crops such as wheat, maize, rice, and soybean are also sensitive to surface ozone pollution, leading to concerns for global food security (Mills et al., 2007). Recent studies estimated that

about 79-121 Tg of crop production was reduced due to ozone pollution in year 2000 alone (Avnery et





al., 2011), and future ozone damage on crops would lead to a 3.6% loss in total crop production under the Intergovernmental Panel on Climate Change (IPCC) RCP8.5 emission scenario (Tai et al., 2014).

The terrestrial biosphere can in turn affect surface ozone levels through surface-atmosphere exchange processes including biogenic VOC emissions, soil $NO_x$ emissions, as well as ozone dry deposition loss (Heald and Geddes, 2016). A number of studies have investigated how surface air quality may respond to perturbations of these processes driven by historical land use change (Fu and Tai, 2015; Val Martin et al., 2015; Fu et al., 2016; Heald and Geddes, 2016) and climate change (Fu and Tai, 2015; Fu et al., 2016). Atmospheric nitrogen deposition, by enhancing plant growth and soil mineral nitrogen content, is

thus expected to modulate the production and loss of surface ozone. In this study, we build an asynchronously coupled modeling system using the GEOS-Chem global atmospheric chemistry model and the Community Land Model (CLM) to quantify the responses of surface ozone air quality to nitrogen deposition since preindustrial times via atmosphere-ecosystem exchange. We examine the individual processes that can be perturbed by nitrogen deposition and then affect surface ozone

concentrations including biogenic VOC emissions, soil $NO_x$ emissions, and ozone dry deposition. To evaluate the relatively importance of nitrogen deposition, we also estimate the surface ozone changes driven by historical climate and land use changes which have been better constrained in recent studies as described above.

**2 Model description**

We combine a chemical transport model (GEOS-Chem) and a global land model (CLM) to investigate to interactions between nitrogen deposition and surface air quality. The interacting processes are given in the schematic diagram in Figure 1 as will be discussed below. Asynchronous coupling of the two models allow us to examine individual processes. We describe in this section the two models, the

asynchronously coupled framework, as well as our model simulations.





## 2.1 The GEOS-Chem chemical transport model

We use the GEOS-Chem global chemical transport model (v9-02; http://www.geos-chem.org) to

characterize the contribution of anthropogenic sources to nitrogen deposition and responses of surface

ozone to changes in vegetation density (as represented by LAI) and soil $NO_x$ emissions as will be

provided by CLM. GEOS-Chem is driven by the MERRA (Modern Era Retrospective-analysis for

Research and Applications) assimilated meteorological data from the NASA Global Modeling and

Assimilation Office (GMAO). We run the GEOS-Chem model at a global horizontal resolution of 2 °

latitude by 2.5 ° longitude, and 47 levels in the vertical.

The GEOS-Chem model has been applied in a number of studies to simulate atmospheric nitrogen

deposition (Zhang et al., 2012; Ellis et al., 2013; Zhao et al., 2015; 2017), surface ozone air quality

(Zhang et al., 2011; 2014; Fu et al., 2015), and recently impacts of land use changes on atmospheric

composition through biosphere-atmosphere exchange processes (Fu and Tai, 2015; Fu et al., 2016;

Geddes et al., 2016; Heald and Geddes, 2016). It includes a detailed simulation of tropospheric

$NO_x$-VOC-$O_3$-aerosol chemistry (Park et al., 2004; Mao et al., 2010). The model wet deposition scheme

including scavenging in convective updraft and large-scale precipitation is described by Liu et al. (2001)

for aerosols, and Mari et al. (2000) and Amos et al. (2012) for soluble gases. The dry deposition

parameterization for gases and aerosols follows a standard big-leaf resistance-in-series model (Wesely,

1989; Zhang et al., 2001). Dry deposition velocities are calculated as a combination of aerodynamics

resistance, boundary-layer resistance, and surface resistance.

We use the global anthropogenic emissions from the EDGAR (the Emissions Database for Global

Atmospheric Research) v4.2 emission inventory (EDGAR, 2015), overwritten by regional inventories

including EMEP (the European Monitoring and Evaluation Program) over Europe (Vestreng and Klein,



2002) and REAS-v2 (the Regional Emission in ASia) over East Asia (Kurokawa et al., 2012) with the

$NH_3$ emission seasonality from Zhao et al. (2015). Natural sources include emissions from biomass

burning, lightning, soil, and the biosphere. Here biomass burning emissions are from the GFED-v3 (the

Global Fire Emissions Database version 3) emission inventory (van der Werf et al., 2010). Lightning

$NO_x$ emissions, as described by Sauvage et al. (2007) and Murray et al. (2012), are calculated using the

cloud top height parameterization of Price and Rind (1992) and vertical distribution of Pickering et al.

(1998), and further spatially redistributed to match satellite observations of lightning flashes.

We implement the following modifications so that GEOS-Chem and CLM have harmonized land

surface properties for simulating surface-atmosphere exchange processes including biogenic VOC

emissions, soil $NO_x$ emissions, and dry deposition. We follow Geddes et al. (2016) and use the 16 plant

function types (PFTs) from CLM in the GEOS-Chem land module. The biogenic VOC emissions in

GEOS-Chem are calculated using the MEGAN v2.1 algorithm based on emission factors of the 16 PFTs

and activity factors accounting for emission responses to soil and meteorological conditions, leaf age,

and LAI (Guenther et al., 2012). The original above-soil $NO_x$ emissions in GEOS-Chem are based on

the empirical parameterization of Hudman et al. (2012). In this study, we calculate the above-soil $NO_x$

emissions in CLM (with improvements as described below and in the Supplement), and archive the

values hourly for input to GEOS-Chem. The canopy reduction and emission pulsing scalars from

Hudman et al. (2012) are applied to estimate the above-canopy $NO_x$ emissions. Furthermore, we have

mapped the 16 CLM PFTs to the deposition surface types of Wesely (1989) following Geddes et al.

(2016) to improve the consistency in dry deposition calculation.

## 2.2 The Community Land Model

We use the Community Land Model (CLM v4.5; Oleson et al., 2013), the land component of the

Community Earth System Model (CESM), to simulate responses of LAI and soil $NO_x$ emissions to



enhanced atmospheric nitrogen deposition from anthropogenic sources. We run the CLM model at the resolution of 2.5 °latitude by 1.9 °longitude, driven by the CRU-NCEP (Climatic Research Unit (CRU)-National Centers for Environmental Prediction (NCEP)) climate forcing dataset (CURNCEP, 2015), which combines the CRU TS3.2 monthly data and the NCEP 6-hour reanalysis data with

additional data over oceans, lakes and Antarctica from Qian et al. (2006). Other model inputs such as initial conditions, surface parameters, and physiological constants are from the CESM input data repository (CESM, 2015).

The CLM model in its active biogeochemistry (BGC) mode simulates detailed terrestrial biogeophysical

and biogeochemical processes such as surface energy fluxes, hydrology, and biogeochemical cycles as described by Oleson et al. (2013). Each grid cell is divided into five land units including vegetation, lake, urban, glacier, and crop. The vegetation-covered areas are further characterized by 16 PFTs. The CLM v4.5 model includes a vertically resolved soil biogeochemistry scheme that considers vertical transport of soil carbon and nitrogen (Koven et al., 2013). In the model, nitrogen input to the soil

mineral nitrogen pool is through atmospheric deposition and biological fixation. The mineral nitrogen can be transformed to organic nitrogen by plant uptake and immobilization, or leave the ecosystem through denitrification, leaching, and other loss processes (Oleson et al., 2013).

CLM v4.5 also includes the Century Nitrogen model of del Grosso et al. (2000), which divides the soil

mineral nitrogen into $NH_4^+$ and $NO_3^-$, and calculates nitrification and denitrification rates accordingly. It allows the model to calculate the $N_2O$ emission fluxes associated with nitrification and denitrification (del Grosso et al., 2000). We further add in the model a parameterization of soil $NO_x$ emissions based on the $NO_x$ and $N_2O$ ratio as described in the Supplement (Section S1). We have implemented some modifications to CLM4.5 so that the simulated soil $NO_x$ emissions are consistent with the GEOS-Chem

scheme (Section S1, Figure S1). These modifications also slightly correct the large CLM



overestimations in the vegetation LAI (Dahlin et al., 2015; Duarte et al., 2017) as shown in Figure S2.

For each CLM simulation in this study, we spin up the model for a thousand years for the soil nitrogen

content to reach equilibrium using the meteorological data of 2006-2010 and present-day or

preindustrial nitrogen deposition fluxes. We use the last five-year results for analysis. We conduct these

idealized near-equilibrium simulations instead of transient simulations because terrestrial ecosystems

respond slowly to the environment changes (Jones et al., 2009). Here we aim to provide a first

quantitative analysis of surface ozone responses, and the near-equilibrium simulations present an

estimate of the long-term influence that might occur in the future (Bala et al., 2013).


### 2.3 Asynchronous coupling and model experiments

As illustrated in Figure 1, reactive nitrogen emitted to the atmosphere by human activities, mainly as

$NH_3$ and $NO_x$, will return to the land surface through wet and dry deposition processes. This deposited

nitrogen will add into the soil mineral nitrogen content, and further enhance plant growth as well as

nitrification and denitrification in the nitrogen limited areas. The influences on surface ozone occur via

three main processes: increasing biogenic VOC emissions while accelerating ozone dry deposition due

to plant growth (as represented by increases in LAI in this study), and perturbing soil $NO_x$ emissions

from the enhanced soil mineral nitrogen pool. Besides $NO_x$, soil mineral nitrogen can also release to the

atmospheric as nitrous acid (HONO), which influences atmospheric oxidative capacity (Su et al., 2011).

Here we do not consider the influence through HONO due to a lack of its emission parameterization

and chemistry simulation in both global models used in this study.

We set up an asynchronously coupled system using GEOS-Chem and CLM to investigate the influences

of nitrogen deposition on surface ozone from the individual processes and the overall effects. We first

calculate the global nitrogen deposition fluxes using the GEOS-Chem model averaged for the years



2006-2010. The simulated nitrogen deposition fluxes are then fed into CLM to compute LAI and soil NO$_x$ emissions. To quantify the perturbations induced by human activities, two sets of GEOS-Chem and CLM simulations are conducted with all anthropogenic emissions turned on or off, representing the consequences of nitrogen deposition at the present-day vs. preindustrial conditions. Anthropogenic

contributions are calculated as the differences between the two simulations. Finally, the CLM-simulated LAI and soil NO$_x$ are returned to GEOS-Chem, which completes the land-atmosphere coupling and allows us to examine the impacts of nitrogen deposition on surface ozone concentrations.

Table 1 summarizes the GEOS-Chem simulations as the final step in this study. These simulations are

conducted with all anthropogenic emissions but with different LAI values and soil NO$_x$ emissions simulated by CLM. The simulation for the present-day condition (Run_all) applies the CLM-simulated present-day LAI and soil NO$_x$ emissions. Its differences from the simulation with natural conditions (Run_nat) estimate the overall effect of anthropogenic nitrogen deposition on surface ozone. By considering the individual processes separately (Run_VOCs, Run_soilnox, and Run_drydep), simulated

ozone differences with Run_nat represent their separated effects.

To evaluate the importance of nitrogen deposition, we put our analyses in the context of comparisons with surface ozone changes driven by historical climate and land use changes. As listed in Table 2, we conduct the GEOS-Chem simulations by using the 1986-1990 MERRA fields (for comparison with the

2006-2010 fields) or the preindustrial land use data (1860 vs. the present-day condition for 2000), generally following the previous work of Fu and Tai (2015) and Heald and Geddes (2016). The impacts of climate change on wildfire emissions (Yue et al., 2015) are not considered here.

## 3 Global emissions and deposition of reactive nitrogen

We first evaluate the model simulation of present-day atmospheric nitrogen deposition at the global




scale. Figure 2 shows the spatial distribution of annual total $NH_3$ and $NO_x$ emissions, and the percentage

contribution from anthropogenic sources averaged over the years 2006-2010. Global total $NH_3$ and $NO_x$

emissions are 62 Tg N $a^{-1}$ and 54 Tg N $a^{-1}$, of which 69% (43 Tg N $a^{-1}$) and 61% (33 Tg N $a^{-1}$) are from

anthropogenic sources. Natural emissions include those from lightning (4.8 Tg N $a^{-1}$ as $NO_x$), biomass

burning (4.9Tg N $a^{-1}$ as $NH_3$ and 6.8 Tg N $a^{-1}$ as $NO_x$), soil (5.6 Tg N $a^{-1}$ as $NH_3$ and 9.4 Tg N $a^{-1}$ as

$NO_x$), and ocean (8.6 Tg N as $NH_3$). About 96% of the anthropogenic emissions are in the Northern

Hemisphere. We will thus focus our analyses on the Northern Hemisphere in the study. East Asia

(especially eastern China and India), Europe, and North America are the major emitting regions with

high ratios of anthropogenic contribution. Over the three regions, total reactive nitrogen emissions reach

more than 100 kg N $ha^{-1}$ $a^{-1}$, 60 kg N ha $^{-1}$ $a^{-1}$ and 50 kg N $ha^{-1}$ $a^{-1}$, and about 75-90% of them are from

anthropogenic sources. Most reactive nitrogen is emitted as $NH_3$ in China and India (62% in China and

71% in India), while $NO_x$ is more abundant in Europe and North America (61% in Europe and 62% in

North America), reflecting their different levels of agricultural activities.

Figure 3 shows GEOS-Chem simulated spatial distributions of annual total (reduced + oxidized)

nitrogen deposition fluxes, and percentage contributions from anthropogenic emissions averaged over

2006-2010. Global total nitrogen deposition is simulated to be 114 Tg N $a^{-1}$, with 59% (67 Tg N $a^{-1}$)

from wet deposition and 41% from dry deposition. 65 Tg N (38 Tg N as $NH_x$ and 27 Tg N as $NO_y$) is

deposited to the continents, and the remaining 49 Tg N is deposited to the ocean. Our results are

comparable with previous global model estimates of Dentener et al. (2006), and more recently,

Lamarque et al. (2013) and Vet et al. (2014). Using an ensemble of 21 global chemical transport models,

Vet et al. (2014) estimated a global total nitrogen deposition of 106 Tg N $a^{-1}$, with 55.6% deposited over

the continental non-coastal areas for 2001. Deposition patterns of reactive nitrogen show a similar

spatial distribution to their emissions due to the short lifetimes. Deposition fluxes reach more than 30 kg

N $ha^{-1}$ $a^{-1}$ in Asia (in particular China and India), and 10 kg N $ha^{-1}$ $a^{-1}$ in Europe and North America, in



agreement with the results of Vet et al. (2014). Anthropogenic emissions contribute 71% of total

nitrogen deposition to the land on a global scale. The anthropogenic contributions are greater than 50%

in the Northern Hemisphere, and reach more than 70% in North America, 80% in Western Europe, and

90% in East Asia.


We compare our simulation with $NH_4^+$ and $NO_3^-$ wet deposition flux measurements available for the

same period of 2006-2010, including measurements from the Acid Deposition Monitoring Network in

East Asia (EANET, 2015) and ten surface monitoring sites in North China from Pan et al. (2012),

European Monitoring and Evaluation Program (EMEP, 2015) in Europe, and National Atmospheric

Deposition Program (NADP, 2015) in North America. There is a lack of direct measurements of dry

deposition fluxes (Vet et al., 2014); however, previous studies have evaluated the GEOS-Chem

simulated nitrogen dry deposition fluxes over the US and China using concentration measurements

from surface sites and satellites, and they showed good agreement (Zhang et al., 2012; Zhao et al.,

2017).


Comparisons of measured vs. simulated wet deposition fluxes over North America, Europe, and Asia

are shown in Figure 3, with the correlation efficient (r) and normalized mean bias (NMB=$\sum_{i=1}^{N}(M_i -$

$O_i) / \sum_{i=1}^{N} O_i$) between the observed (O) and modeled (M) values over the $N$ sites computed. Over the

three high nitrogen depositing continents, comparisons generally show high correlation coefficients (r =

0.50-0.86) and low biases for both $NH_4^+$ and $NO_3^-$ wet deposition, except for biases of -21% for $NH_4^+$

wet deposition over Europe and -23% for $NO_3^-$ over East Asia. The high negative biases are likely due

to the difficulty of simulating very high deposition fluxes measured at urban sites as suggested by Zhao

et al. (2017) that evaluated GEOS-Chem-estimated nitrogen deposition over China at a finer horizontal

resolution. Globally, the model is able to capture the magnitudes and spatial distribution of observations

with high correlation coefficients of 0.86 for $NH_4^+$ and 0.70 for $NO_3^-$ and small biases of -5% for $NH_4^+$



and -8% for $NO_3^-$, providing credence to the model simulation of present-day atmospheric nitrogen deposition.

## 4 Impact of anthropogenic nitrogen deposition on land properties

**4.1 Changes in vegetation LAI and subsequent responses**

We discuss in this section the changes in ecosystem structure in terms of foliage density driven by present-day anthropogenic nitrogen deposition as simulated by CLM. Figure 4 shows the simulated present-day spatial distribution of vegetation LAI, and the perturbations due to anthropogenic nitrogen deposition calculated as the differences between CLM simulations forced by total vs. natural-only

nitrogen deposition. Vegetation growth is limited by nitrogen supply over most of the globe. We find that anthropogenic nitrogen deposition enhances global net primary production (NPP) by 3.7 Pg C a$^{-1}$, increasing LAI over those nitrogen-limited areas. Our estimated global NPP increase is consistent with Bala et al. (2013) that used an earlier version of CLM (CLM4.0) and showed that doubling (quadrupling) nitrogen deposition from the preindustrial level would increase global NPP by 2.6 Pg C a$^{-1}$ (6.8 Pg C

a$^{-1}$). As shown in Figure 4, LAI values increase by 0.1-0.7 cm$^2$ cm$^{-2}$, 0.1-0.9 cm$^2$ cm$^{-2}$, and greater than 1.0 cm$^2$ cm$^{-2}$ due to anthropogenic nitrogen deposition over the three hotspots of nitrogen deposition, North America, Europe, and East Asia, respectively. The high increases over southeastern China may also partly reflect the LAI overestimation in CLM (Supplementary Figure S2); we will discuss the associated uncertainties in the next section.


Enhancement in LAI can subsequently lead to higher biogenic VOC emissions, and also higher ozone dry deposition velocities by lowering surface resistance. As shown in Figure 4, GEOS-Chem simulates a global total isoprene emission of 474 Tg a$^{-1}$ for the present-day condition, and anthropogenic nitrogen deposition contributes about 6.6 Tg a$^{-1}$ (1.4 %) from the LAI enhancement. Isoprene emissions are more

sensitive to LAI changes at lower LAI areas due to suppression of sunlight from dense leaves. Thus





emissions over southeastern China do not show large increases despite significant LAI enhancement
(0.8-1.0 $cm^2$ $cm^{-2}$), while smaller LAI changes (0.2-0.6 $cm^2$ $cm^{-2}$) over regions such as India and
southeastern Brazil lead to distinct increases in isoprene emissions up to 10-15%. As for dry deposition,
the deposition velocities for ozone tend to increase with increasing LAI. We estimate that anthropogenic

nitrogen deposition increases ozone dry deposition velocity by about 0.02 cm $s^{-1}$ (~8%) over eastern US
and Western Europe, and 0.04 cm $s^{-1}$ (10%) over eastern and southern Asia.

## 4.2 Changes in soil $NO_x$ emissions

Figure 5 shows how addition of deposited anthropogenic nitrogen to the soil mineral nitrogen pool

could perturb soil $NO_x$ emissions. As described above, we calculate the above-soil $NO_x$ emissions in
CLM using a scaling parameterization with respect to $N_2O$ emission fluxes associated with nitrification
and denitrification. Our CLM model results estimate that anthropogenic nitrogen deposition contributes
to global emissions of 45.4 Tg N $a^{-1}$ as $N_2$, 1.32 Tg N as $N_2O$, and 2.6 Tg N as $NO_x$ above soil. Zaehle
et al. (2011) previously estimated that global $N_2O$ emissions were increased by 0.8 Tg N $a^{-1}$ from 1860

to 2005 due to atmospheric nitrogen deposition using transient simulations of a terrestrial land model.
Our estimate (1.32 Tg N-$N_2O$) is reasonably higher considering we use the near-equilibrium
simulations.

We estimate the present-day global above-canopy $NO_x$ emissions to be 9.4 Tg N $a^{-1}$ (Figure 5), and they

are in good agreement with the results estimated by the GEOS-Chem soil $NO_x$ scheme of Hudman et al.
(2012) for the same period in terms of both the global magnitude (9.3 Tg N $a^{-1}$) and spatial distribution
(Supplementary Figure S1). 1.9 Tg N $a^{-1}$ of the above-canopy $NO_x$ emissions are contributed by
addition of deposited anthropogenic nitrogen, and 46% of the increased emissions occur over China and
India. As shown in Figure 5, anthropogenic nitrogen deposition can lead to significant increases in the

soil $NO_x$ emissions especially in the Northern Hemisphere. These increases account for 30-70% of local

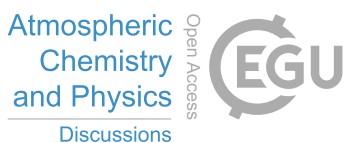

soil $NO_x$ emissions over regions of China, India, the US, and Europe. There is also a strong seasonality in the enhanced soil $NO_x$ emissions since nitrification and denitrification rates are highly dependent on surface temperature. We find that 41% (0.77 Tg N) of the emission enhancement occurs in Jun-July-August, and only 13% in December-January-February.


## 5 Responses of surface ozone pollution

### 5.1 Surface ozone concentration

We now examine the changes in surface ozone air quality as driven by the overall effect of anthropogenic nitrogen deposition, as well as the individual processes of dry deposition, biogenic VOC

and soil $NO_x$ emissions. We use the metric of daytime (08:00-18:00 local time) mean surface ozone concentration. Figure 6 and 7 show the resulting surface ozone changes in the Northern Hemisphere averaged over summer (June-July-August) and spring (March-April-May), respectively. We find overall increases in the surface ozone concentration over the globe except for some regions with high $NO_x$ emissions. In June-July-August, the mean surface ozone changes are generally within ±3 ppbv, with

about 1 ppbv increases over the southwestern US and Northern Europe, nearly 2 ppbv over India, north-central China, and northern African grassland; while about 0.5 ppbv decreases in the eastern US, and 1 ppbv decreases over North China and Western Europe. Similar patterns are found for spring (March-April-May) (Figure 7), but changes are weaker than summer.

The overall impacts of nitrogen deposition on surface ozone are buffered from their effects through individual processes. Figure 6 and 7 also show the separated effects from changes in dry deposition, biogenic VOC emissions, and soil $NO_x$ emissions. Increases in vegetation LAI tend to increase surface ozone concentrations due to higher biogenic VOC emissions, but are largely offset by increases in the ozone loss through higher dry deposition velocities. The net effects of two depend on the LAI values

and the relative changes. For example, India as one of the regions with the largest relative changes in





LAI shows higher ozone changes driven by biogenic VOC emissions than the decreases from dry

deposition. Meanwhile, the ozone responses to soil $NO_x$ emissions are nonlinear depending on whether

the area is $NO_x$-limited or $NO_x$-saturated. As shown in Figure 6f, soil $NO_x$ emission enhancements

generally increase summer mean surface ozone concentrations in the Northern Hemisphere expect for

North China, where anthropogenic $NO_x$ emissions are particularly large and ozone production is limited

by VOC emissions as reported in recent studies (Tang et al., 2012).

One of the largest uncertainties arises from the CLM overestimation of vegetation LAI (Supplementary

Figure S2). To test this uncertainty, we have conducted another set of simulations in which the

GEOS-Chem model simulations use observed LAI from the MODIS satellite instrument, and the

CLM-simulated present-day vs. natural LAI enhancement ratios are applied to adjust MODIS LAI to

examine the contribution of anthropogenic nitrogen deposition. The resulting impacts of anthropogenic

nitrogen deposition on biogenic isoprene emissions, ozone dry deposition velocities, and summer mean

surface ozone concentrations are shown in Supplementary Figure S3. Using adjusted MODIS LAI

would lead to larger increases in biogenic isoprene emissions from anthropogenic nitrogen deposition

(globally 8.2 Tg $a^{-1}$ with the MODIS LAI vs. 5.6 Tg $a^{-1}$ with the CLM LAI for the year 2009), and

weaker increases in dry deposition velocity. As for summer mean surface ozone, we find that the

differences are minor over the globe except for southeastern China where the largest LAI overestimate

in CLM occurs. Changes in summer mean surface ozone due to anthropogenic nitrogen deposition are

around 0.1-2.0 ppbv with the adjusted MODIS LAI but are overall negative (up to -1.0 ppbv) with the

CLM LAI over this region, reflecting the combined effect of enhanced biogenic VOC emissions and

reduced ozone dry deposition loss with lower LAI.

**5.2 Comparisons with climate and land use driven surface ozone changes**

We also show in Figure 6 and 7 surface ozone changes driven by historical climate (1986-1990 vs.



2006-2010) and land use (1860 vs. 2000) changes as simulated by the GEOS-Chem model. The changes

for ozone dry deposition velocity, biogenic isoprene, and soil $NO_x$ emissions are included in the

Supplement (Figure S4). Surface ozone changes from the 20-year climate change show a large spatial

variability with more than ±10 ppbv concentration changes in both spring and summer. The large

variations are mainly driven by changes in surface temperature and other meteorological variables as

found in previous studies (Camalier et al., 2007; Jacob et al., 2009; Doherty et al., 2013). The large

ozone concentration increases over northern Eurasia and Africa are associated with higher temperature

in 2006-2010 relative to 1986-1990, which leads to higher biogenic VOC emissions (Figure S4) and

stronger ozone photochemical production rates. Meanwhile higher temperature decreases surface ozone

over remote regions (ocean and deserts) due to stronger ozone loss and less PAN transported from

source regions (Doherty et al., 2013). Part of the ozone differences are also associated with changes in

ozone dry deposition velocities and soil $NO_x$ emissions as driven by changes in temperature and

planetary boundary layer (PBL) height (Figure S4 and S5).

The historical land use change has led to decreases in surface ozone concentrations up to 5-7 ppbv for

the summer mean over most regions except for some areas in Western Europe, North China, and central

Africa where there are slight increases. These results are consistent with the recent work of Heald and

Geddes (2016) that investigated the impacts of changes in land types and agricultural activities on

surface air quality. The land-use-induced surface ozone changes are largely caused by a shift of forest

trees with high biogenic emission factors to grasslands and croplands with low emission factors from

1860 to 2000. This shift in land types has also led to changes in ozone dry deposition velocity by up to

10% due to the combined impacts of LAI changes, cropland expansion (enhancing ozone vegetation

uptake), and deforestation (decreasing ozone dry deposition velocity) (Heald and Geddes, 2016).

Compared with the impacts from climate change, surface ozone changes induced by anthropogenic



nitrogen deposition (±3 ppbv) are smaller on a global scale, but can be rather important at the local and regional scales. The anthropogenic nitrogen deposition induced ozone changes are usually 10% of those induced by climate change at low and middle latitudes but reach about 50% at high latitudes (e.g., Canada and Northern Europe). These values are also comparable to impacts from land use changes over

regions such as the western US and India, where nearly all surface ozone concentration decreases due to historical land use changes are compensated by the increases caused by anthropogenic nitrogen deposition (0.3-1.5 ppbv over the western US and 0.5-2.3 ppbv over India).

## 6 Conclusions

In this study we present an exploratory study aiming to quantify the influences of anthropogenic nitrogen deposition on surface ozone air quality by using the GEOS-Chem chemical transport model asynchronously coupled with the CLM land model. Increased atmospheric nitrogen deposition from human activities can modulate plant growth and mineral nitrogen content in soil, and further affect atmospheric composition through surface-atmosphere exchange processes. We consider here three

processes including biogenic VOC emissions, ozone dry deposition, and soil $NO_x$ emissions. A combination of GEOS-Chem and CLM allows us to investigate how these processes influence surface ozone and how anthropogenic nitrogen deposition perturbs them.

We simulate in GEOS-Chem global atmospheric nitrogen deposition fluxes for the present-day and the
preindustrial (natural emissions only) conditions, and then conduct near-equilibrium CLM simulations with these fluxes to estimate terrestrial vegetation LAI, soil $NO_x$ emissions, and their changes due to anthropogenic nitrogen deposition. The present-day (2006-2010) nitrogen deposition is estimated to be 114 Tg N $a^{-1}$ with 57% (65 Tg N $a^{-1}$) deposited to the land, consistent with available measurements of wet deposition fluxes. Anthropogenic sources contribute 71% of the nitrogen deposition to the land on
the global scale, and 70%-90% over the Northern Hemisphere continents. We find that anthropogenic



nitrogen deposition leads to large-scale increases in LAI as well as soil $NO_x$ emissions over the globe.
The contributions from anthropogenic nitrogen deposition are particularly high over North America,
Europe, and East Asia, with local values of 5%-30% for LAI and 20%-70% for present-day soil $NO_x$
emissions.


Surface ozone changes driven by anthropogenic nitrogen deposition are then identified in additional
GEOS-Chem simulations with CLM-simulated LAI and soil $NO_x$ emissions. We find that the LAI
enhancement due to anthropogenic nitrogen deposition can increase biogenic VOC emissions (e.g., a
6.6 Tg increase in isoprene emissions), but also lead to higher ozone dry deposition velocities (1%-15%
increases over the Northern Hemisphere continents). Surface ozone changes due to the two processes
are largely offset. Anthropogenic nitrogen deposition also leads to general increases in soil $NO_x$
emissions that increase the seasonal mean surface ozone concentrations over the globe except for North
China where ozone production is found to be $NO_x$ saturated. We find that the net effects of
anthropogenic nitrogen deposition lead to summer mean surface ozone increases of 1 ppbv over the
southwestern US, 2 ppbv over India, north-central China; while decreases of 0.5 ppbv in the eastern US,
and 1 ppbv over North China and Western Europe.

To assess the importance of deposited anthropogenic nitrogen influences, we also estimate surface
ozone changes driven by the past 20-year climate change (from 1986-1990 to 2006-2010) and historical
land use change (from 1860 to 2000). The 20-year climate change has led to large changes in the
seasonal mean surface ozone concentration (±10 ppbv), mainly driven by changes in temperature and
other meteorological variables, while the historical land use change induces decreases of summer mean
surface ozone by up to 5-7 ppbv in the Northern Hemisphere due to deforestation and cropland
expansion as discussed in recent studies (Fu and Tai, 2015; Heald and Geddes, 2016). Compared with
those changes, we find that the influences of anthropogenic nitrogen deposition can be comparable at



regional scales. In particular, they may largely offset the surface ozone reduction due to historical land use change over the Northern Hemisphere continents.

While our study points out that anthropogenic nitrogen deposition can be important in modulating the

surface ozone air quality, it shall be acknowledged that considerable uncertainties still exist. The estimated surface ozone responses rely heavily on the parameterizations of surface-atmosphere exchange processes. Using different parameterizations with different meteorological data, large ranges have been found for the estimates of biogenic emissions (Guenther et al., 2012; Henrot et al., 2016), soil $NO_x$ emissions (Hudman et al., 2012), and ozone dry deposition velocities (Hardacre et al., 2015).

Future work is needed to reconcile them especially in light of observations and understand the uncertainty ranges. The near-equilibrium CLM simulations applied in this study also imply that our estimates represent a long-term, steady-state impact, and may represent quite difference results from the transient responses to actual perturbations of the terrestrial nitrogen cycle over centurial timescales. In addition, previous studies have shown that nitrogen deposition can lead to reduction of plant diversity

(Sutton et al., 2014). This is not considered in our study since we use prescribed, constant PFT distribution, soil types and soil pH. All the possible uncertainties reflect the complexity in the biosphere-atmosphere interactions and feedbacks, and require future efforts in better characterizing these exchange processes in finer integrated models such as Earth system models.

**Data availability**

The datasets including measurements and model simulations can be accessed from websites listed in the references or by contacting the corresponding author (Lin Zhang; zhanglg@pku.edu.cn).

**Acknowledgments**

This work was supported by China's National Basic Research Program (2014CB441303), and the



National Natural Science Foundation of China (41205103, 41475112, and 41405144). The collaboration was also supported by the General Research Fund (project #: 14323116) of the Research Grants Council of Hong Kong given to Amos P. K. Tai.

**A description of the soil NOₓ emission parameterization and 5 Figures are included in the supplement related to this article.**

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



**Tables**

**Table 1.** GEOS-Chem simulations to quantify surface ozone response to nitrogen deposition from each

process and the net effect

|  | Run_all | Run_VOCs | Run_soilnox | Run_drydep | Run_nat |
|---|---|---|---|---|---|
| **Biogenic VOCs** | All[1] | All | Nat[2] | Nat | Nat |
| **Soil NO$_x$** | All | Nat | All | Nat | Nat |
| **Dry deposition** | All | Nat | Nat | All | Nat |

[1]In the table All represents the use of CLM outputs simulated with the present-day atmospheric nitrogen deposition,
and Nat represents the use of CLM outputs with natural nitrogen deposition alone.

**Table 2.** GEOS-Chem simulations with the input data time listed to quantify surface ozone changes

driven by historical climate and land use changes

|  | Run_std | Run_met | Run_land |
|---|---|---|---|
| **Land use** | 2000 | 2000 | 1860 |
| **Meteorology** | 2006-2010 | 1986-1990 | 2006-2010 |




## Figures


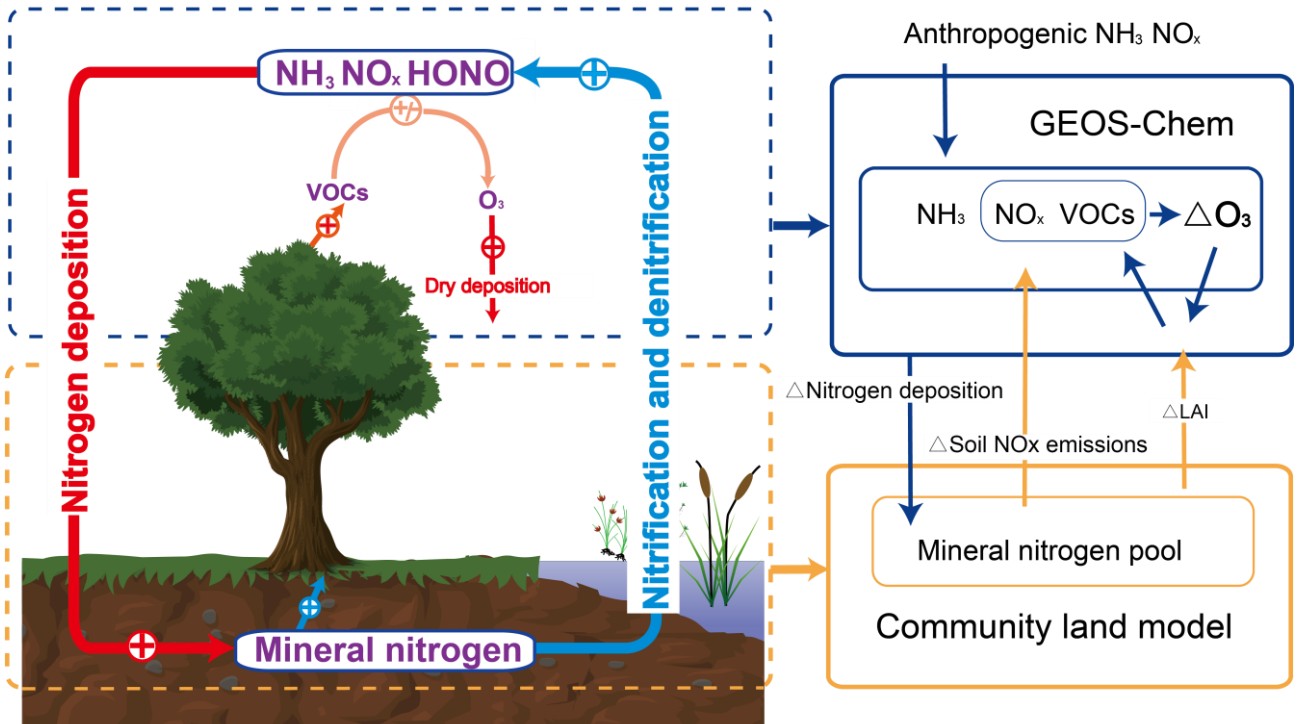

**Figure 1**. The schematic diagram and flowchart of the land-atmosphere asynchronously coupled system for the study.






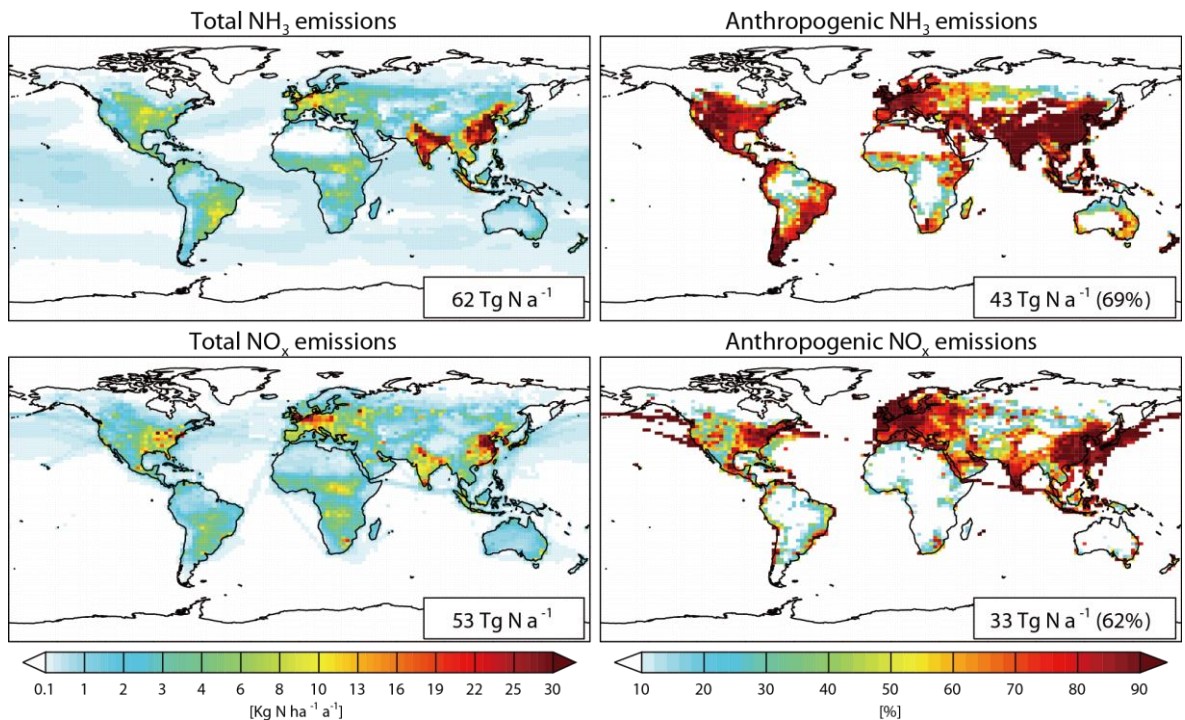

**Figure 2**. Spatial distribution of total NH₃ (top panels) and NOₓ (bottom panels) emissions (left panels)
and percentage contribution from anthropogenic sources (right panels) averaged for 2006-2010. Annual
global total values are shown inset.








**Figure 3**. Top panels show total nitrogen deposition fluxes (left) and contributions from anthropogenic sources (right). Annual total deposition values to land are shown inset. Bottom panels compare the simulated $NH_4^+$ (left) and $NO_3^-$ (right) wet deposition fluxes with an ensemble of surface measurements over North America, Europe, and Asia as described in the text. The comparison scatter-plots are over-plotted with correlation coefficients (r) and normalized mean biases (b) also shown inset.







**Figure 4**. Leaf area index (top panels), biogenic isoprene emission (middle panels with annual total emissions shown inset), and dry deposition velocity for ozone (bottom panels) as simulated by the asynchronously coupled modeling system. The left panels represent the present-day conditions, and the right panels show perturbations as could be driven by human-induced atmospheric nitrogen deposition.





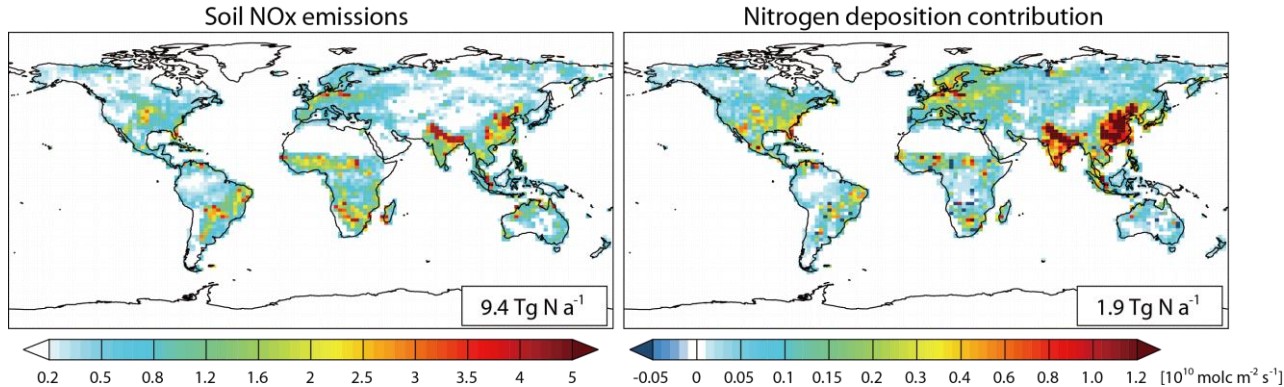

**Figure 5**. Present-day soil $NO_x$ emissions (left) and contributions from anthropogenic nitrogen deposition (right). Annual totals are shown inset.



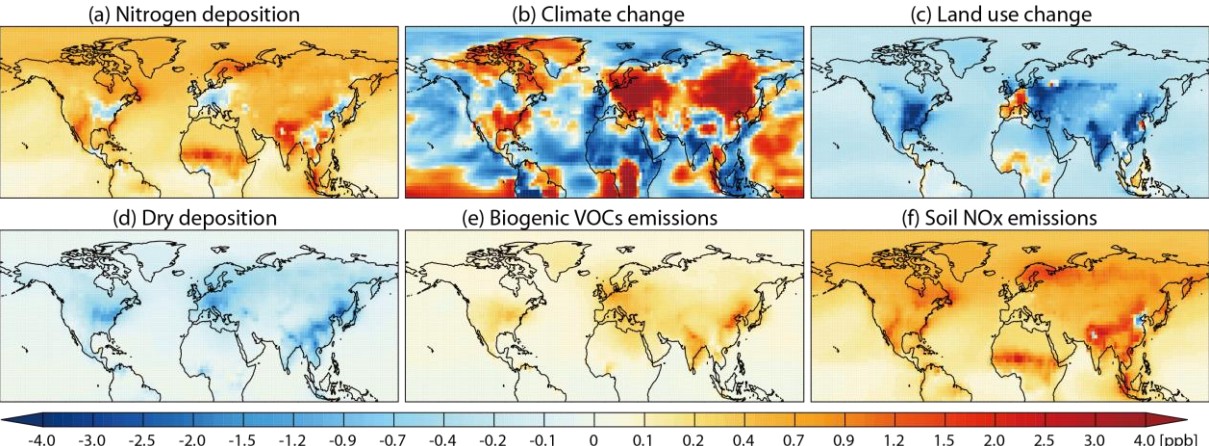

**Figure 6**. Changes in mean surface ozone concentration for June-July-August driven by anthropogenic nitrogen deposition (top-left panel), changes in climate (2006-2010 vs. 1986-1990; top-middle panel) and land use (present-day vs. 1860 conditions; top-right panel). Model simulations are described in the text. Bottom panels separate the anthropogenic nitrogen deposition-induced ozone changes into those due to three processes: changes in dry deposition velocity, biogenic VOC emissions, and soil NO$_x$ emissions.

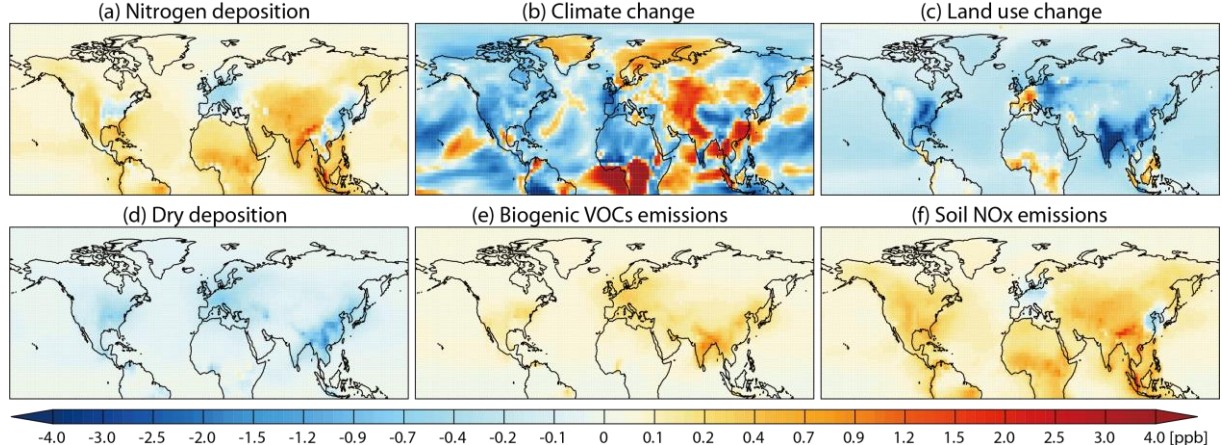

**Figure 7**. Same as Figure 6 but for March-April-May.