# Peer review of "Responses of surface ozone air quality to anthropogenic nitrogen deposition in the Northern Hemisphere"

_Atmospheric Chemistry and Physics, 2017_

## Referee Comment (RC1) · Anonymous Referee #1 · 11 May 2017

In this manuscipt, Zhao et al. present the results of an exploratory modeling study that quantifies the potential impact of anthropogenic nitrogen deposition on ozone air quality. The GEOS-Chem chemical transport model is used to derive nitrogen deposition fields (with and without anthropogenic emissions) that are used in separate Community Land Model simulations in order to derive contrasting global vegetation properties and soil NOx emissions. These are in turn used in GEOS-Chem to simulate impacts on surface ozone concentrations. The authors find that anthropogenic nitrogen deposition can increase surface ozone by enhancing biogenic VOC, but also decrease surface ozone by enhancing dry deposition velocities. Changes in O3 resulting from increased soil NOx emissions are also spatially heterogeneous. The simulated effects on O3 from

anthropogenic nitrogen deposition are comparable to predicted impacts resulting from land use change alone.

In my opinion, this manuscript is novel, logically presented, and mostly well-written. By asynchronously coupling the Community Land Model with the GEOS-Chem chemical transport model, the authors present an enlightening approach to isolating specific land-system processes on atmospheric chemistry. The results suggest that a more refined consideration of biosphere-atmosphere coupling can have appreciable impacts on atmospheric chemistry. Like any "exploratory" modeling study, it is difficult to evaluate the implications directly with observations, but I believe this manuscript points the research community in a constructive direction. This work will surely be of interest to the Atmospheric Chemistry and Physics audience. I have only minor concerns and technical corrections to suggest.

One concern I have is that the approach seems like it would be very difficult for others to reproduce, given the variety of simulations, the dependence on land and atmospheric data products, and the asynchronous coupling. The authors appear to try and address this challenge, offering to provide the measurements and model simulations upon request. I might encourage the authors to provide separately the N-deposition fields, soil-NOx emission fields, and land cover inputs in order to facilitate potential intercomparison studies with other models.

It is also regrettable to me that the changes to the CLM relating to soil NOx emissions, NH3 volatilization, and N uptake, are relegated to the Supplementary Information. I believe these modifications could be of great interest (and debate?) to both model communities, and might stimulate constructive discussions about model development. However, given that the present manuscript already presents a substantial amount of material (and given that model development is somewhat outside the scope of ACP), I understand the authors' motivations for doing so. Is it possible there is room for an Appendix to the article instead (and the figures could be retained in the Supplemental Information)? I leave this to the authors' discretion.

Technical Corrections:

line 66: remove "been"

line 91: replace "relatively" with "relative"

Section 2.3: It wasn't explicitly clear to me until later in the manuscript (actually, the second last sentence) that prescribed land cover/vegetation PFT/soil types in the CLM simulations with- and without anthropogenic N deposition are constant (not dynamically changing over time or between simulations). I believe this should be clarified here, since the impacts of land use change are addressed in a separate investigation. Also, what is the source of the prescribed land/vegetation cover? What time period does it represent?

Figure 3: I found the top right panel a bit confusing. This plot shows the percentage contributions to N-dep from anthropogenic emissions. Can you lay out exactly what model(s) output subtraction you are carrying out here?

line 289-290: The two numbers are both given units of Pg C a-1.

line 354: replace "expect" with "except"?

line 460: "especially in light of observations" – What observations are the authors referring to?

line 463: replace "difference" with "different".

Table 1: There is a superscript "2" under Run_soilnox, but no footnote associated with it. Furthermore, I think it could be clarified further (in the footnote or in the heading) that the GEOS-Chem simulations that address the N-deposition impact on O3 are run with present-day anthropogenic emissions. (I.e. There are essentially two families of final simulations: GEOS-Chem + anthro emissions + plant cover driven by natural N deposition _VS._ GEOS-Chem + anthro emissions + plant cover driven by natural and anthropogenic N deposition.)

---

## Referee Comment (RC2) · Anonymous Referee #2 · 30 May 2017

This manuscript presents a modeling study on the effects of anthropogenic nitrogen deposition on ozone pollution through surface-atmosphere exchanges. The Authors combine GEOS-Chem with CLM to produce an interesting, smart modeling experiment to study several processes (e.g. soil NOx, biogenic VOC, LAI, dry deposition velocity, etc) that affect surface ozone. They find that enhance atmospheric N deposition increases surface ozone by changes in biogenic VOC emissions and dry deposition velocities. Enhanced atmospheric N deposition also increases soil NOx emissions, but the effect on surface ozone is more scattered. The resulted increases in ozone are comparable to changes in climate and land use alone, as determined by previous studies. This study highlights the importance of considering biosphere-atmosphere feedbacks in future

airquality predictions.

The effect of human activities on atmospheric N deposition and further consequences to ecosystems and air quality is an important topic and results from this work are of relevant interest for the ACP readers. The manuscript is of very good quality, well written and organized. I did not find any major concern with the analysis and manuscript in general, and consider this article adequate for publication. I have added a few comments and editorial corrections below, and I hope the Authors consider in the revision of their manuscript.

Main Text

Line 153 Typo, ‘CRUNCEP’ not ‘CURNCEP’

Supplementary Material

Line 21. In ‘RNOx:N2O’ equation. Is ‘ATAN’ the Arctangent?

Line 61 Typo, ‘Hanes Woolf’ Mechanism,

Line 67 Odd sentence, “f(Tsoil) represents the limitation of soil temperature on plant nitrogen uptake that we apply the same function as soil decomposition and nitrification in CLM”.

Line 75 How does the modifications in CLM (NH3 volatilization, N update and soil NOx) correct LAI? Do all these modifications contribute the same, or is it mainly because the N uptake by the plants? May you please elaborate?

---

## Author Comment (AC1) · 12 Jul 2017

**Comment:** In this manuscipt, Zhao et al. present the results of an exploratory modeling study that quantifies the potential impact of anthropogenic nitrogen deposition on ozone air quality. The GEOS-Chem chemical transport model is used to derive nitrogen deposition fields (with and without anthropogenic emissions) that are used in separate Community Land Model simulations in order to derive contrasting global vegetation properties and soil NOx emissions. These are in turn used in GEOS-Chem to simulate impacts on surface ozone concentrations. The authors find that anthropogenic nitrogen deposition can increase surface ozone by enhancing biogenic VOC, but also decrease

surface ozone by enhancing dry deposition velocities. Changes in O3 resulting from increased soil NOx emissions are also spatially heterogeneous. The simulated effects on O3 from anthropogenic nitrogen deposition are comparable to predicted impacts resulting from land use change alone.

In my opinion, this manuscript is novel, logically presented, and mostly well-written. By asynchronously coupling the Community Land Model with the GEOS-Chem chemical transport model, the authors present an enlightening approach to isolating specific land-system processes on atmospheric chemistry. The results suggest that a more refined consideration of biosphere-atmosphere coupling can have appreciable impacts on atmospheric chemistry. Like any "exploratory" modeling study, it is difficult to evaluate the implications directly with observations, but I believe this manuscript points the research community in a constructive direction. This work will surely be of interest to the Atmospheric Chemistry and Physics audience. I have only minor concerns and technical corrections to suggest.

**Response:** We thank the reviewer for the helpful comments. All of them have been addressed in the revised manuscript. Please see our itemized responses below.

**Comment:** One concern I have is that the approach seems like it would be very difficult for others to reproduce, given the variety of simulations, the dependence on land and atmospheric data products, and the asynchronous coupling. The authors appear to try and address this challenge, offering to provide the measurements and model simulations upon request. I might encourage the authors to provide separately the N-deposition fields, soil-NOx emission fields, and land cover inputs in order to facilitate potential intercomparison studies with other models.

**Response:** We agree with the suggestion. We have now put the model simulations of nitrogen deposition flux, soil-NOx emission, and leaf area index on the webpage (http://www.phy.pku.edu.cn/~atmoscc/data/acp-2017-242-data.html)

We also state in the "Data availability" section: "The datasets including measurements

and model simulations can be accessed from websites listed in the references, downloaded from the webpage (http://www.phy.pku.edu.cn/~atmoscc/data/acp-2017-242-data.html), or by contacting the corresponding author (Lin Zhang; zhanglg@pku.edu.cn)"

**Comment:** It is also regrettable to me that the changes to the CLM relating to soil NOx emissions, NH3 volatilization, and N uptake, are relegated to the Supplementary Information. I believe these modifications could be of great interest (and debate?) to both model communities, and might stimulate constructive discussions about model development. However, given that the present manuscript already presents a substantial amount of material (and given that model development is somewhat outside the scope of ACP), I understand the authors' motivations for doing so. Is it possible there is room for an Appendix to the article instead (and the figures could be retained in the Supplemental Information)? I leave this to the authors' discretion.
**Response:** We also think that it is a good idea to use Appendix. As suggested, we now move the description of CLM modifications to the Appendix, and keep the relevant figures in the Supplement.

**Comment:** line 66: remove "been"
**Response:** Changed as suggested.

**Comment:** line 91: replace "relatively" with "relative"
**Response:** Changed as suggested.

**Comment:** It wasn't explicitly clear to me until later in the manuscript (actually, the second last sentence) that prescribed land cover/vegetation PFT/soil types in the CLM simulations with- and without anthropogenic N deposition are constant (not dynamically changing over time or between simulations). I believe this should be clarified here, since the impacts of land use change are addressed in a separate

investigation. Also, what is the source of the prescribed land/vegetation cover? What time period does it represent?

**Response:** We now state in Section 2.2 (the Community Land Model): "The vegetation-covered areas are further characterized by 16 PFTs, which are derived from MODIS observations to represent the present-day condition (Lawrence and Chase, 2007)." And "The CLM simulations use prescribed, constant PFT distribution and soil types. We will investigate the influences of land use change on surface ozone by a separate GEOS-Chem simulation as described below."

Added reference: Lawrence, P. J., and Chase, T. N.: Representing a new MODIS consistent land surface in the Community Land Model (CLM 3.0), J. Geophys. Res.-Biogeo., 112, G01023, 10.1029/2006JG000168, 2007.

**Comment:** Figure 3: I found the top right panel a bit confusing. This plot shows the percentage contributions to N-dep from anthropogenic emissions. Can you lay out exactly what model(s) output subtraction you are carrying out here?
**Response:** We now state in the caption of Figure 3: "contributions from anthropogenic sources estimated as percentage changes in the GEOS-Chem simulation with all anthropogenic emissions turned off relative to the simulation with anthropogenic emissions turned on (right)".

**Comment:** line 289-290: The two numbers are both given units of Pg C a-1.
**Response:** Yes, we now state "increase global NPP by 2.6 (6.8) Pg C a-1".

**Comment:** line 354: replace "expect" with "except"?
**Response:** Changed as suggested.

**Comment:** line 460: "especially in light of observations" – What observations are the authors referring to?

**Response:** We now state: "Future work is needed to reconcile them especially in light of more observations of these emission and deposition fluxes and understand the uncertainty ranges."

**Comment:** line 463: replace "difference" with "different".
**Response:** Changed as suggested.

**Comment:** Table 1: There is a superscript "2" under Run_soilnox, but no footnote associated with it. Furthermore, I think it could be clarified further (in the footnote or in the heading) that the GEOS-Chem simulations that address the N-deposition impact on O3 are run with present-day anthropogenic emissions. (I.e. There are essentially two families of final simulations: GEOS-Chem + anthro emissions + plant cover driven by natural N deposition _VS._ GEOS-Chem + anthro emissions + plant cover driven by natural and anthropogenic N deposition.)
**Response:** Thanks for pointing it out. We have clarified this by merging the footnotes into one for the table title: "1 In the table All represents the use of CLM outputs simulated with the present-day atmospheric nitrogen deposition, and Nat represents the use of CLM outputs with natural nitrogen deposition alone. All GEOS-Chem simulations listed in the table are conducted with present-day anthropogenic and natural emissions turned on."

---

## Author Comment (AC2) · 12 Jul 2017

**Comment:** This manuscript presents a modeling study on the effects of anthropogenic nitrogen deposition on ozone pollution through surface-atmosphere exchanges. The Authors combine GEOS-Chem with CLM to produce an interesting, smart modeling experiment to study several processes (e.g. soil NOx, biogenic VOC, LAI, dry deposition velocity, etc) that affect surface ozone. They find that enhance atmospheric N deposition increases surface ozone by changes in biogenic VOC emissions and dry deposition velocities. Enhanced atmospheric N deposition also increases soil NOx emissions, but the effect on surface ozone is more scattered. The resulted

increases in ozone are comparable to changes in climate and land use alone, as determined by previous studies. This study highlights the importance of considering biosphere-atmosphere feedbacks in future air quality predictions. The effect of human activities on atmospheric N deposition and further consequences to ecosystems and air quality is an important topic and results from this work are of relevant interest for the ACP readers. The manuscript is of very good quality, well written and organized. I did not find any major concern with the analysis and manuscript in general, and consider this article adequate for publication. I have added a few comments and editorial corrections below, and I hope the Authors consider in the revision of their manuscript.

**Response:** We thank the reviewer for the helpful comments. All of them have been addressed in the revised manuscript. Please see our itemized responses below.

Main Text
**Comment:** Line 153 Typo, 'CRUNCEP' not 'CURNCEP'
**Response:** Changed as suggested.

Supplementary Material
**Comment:** Line 21. In 'RNOx:N2O' equation. Is 'ATAN' the Arctangent?
**Response:** Yes, we now state: "the NOx over N2O emission ratio, which varies with the gas diffusivity (D/D0) as described by the arctangent (ATAN) function (Parton et al., 2001)".

**Comment:** Line 61 Typo, 'Hanes Woolf' Mechanism,
**Response:** Thanks for pointing it out. The typo is now corrected.

**Comment:** Line 67 Odd sentence, "f(Tsoil) represents the limitation of soil temperature on plant nitrogen uptake that we apply the same function as soil decomposition and nitrification in CLM".

**Response:** We change the sentence to: "f(Tsoil) represents a function of limitation of soil temperature on plant nitrogen uptake as described in Thomas et al. (2013)".

**Comment:** Line 75 How does the modifications in CLM (NH3 volatilization, N update and soil NOx) correct LAI? Do all these modifications contribute the same, or is it mainly because the N uptake by the plants? May you please elaborate?
**Response:** The correction to LAI is mainly from updates on plant nitrogen uptake. We now add in the setion2.2: "These modifications also slightly correct the large CLM overestimations in the vegetation LAI (Dahlin et al., 2015; Duarte et al., 2017) (Figure S2) mainly due to reduced nitrogen uptake by plant in our model."